# Position: Limitations of LLMs Can Be Overcome by Carefully Designed Multi-Agent Collaboration

## Abstract

**Position Statement**:

Current Large Language Models (LLMs) face three fundamental limitations: (1) reliance on pattern matching rather than deliberate reasoning, (2) inability to self-validate their output, similar to Gödel's incompleteness constraints, and (3) inconsistent constraint management in planning tasks. These deficiencies prevent LLMs from achieving system-2 level reasoning and planning.

We introduce Multi-Agent Collaborative Intelligence (MACI), a structured framework designed to overcome these challenges through meta-planning and distributed validation. MACI comprises three key components: (1) a *meta-planner* (MP) that formulates and refines all task roles and constraints while generating a dependency graph augmented with common-sense reasoning; (2) a *collection of specialized agents* to facilitate domain-specific planning and task execution; and (3) a *run-time monitor* that dynamically adjusts plans as needed. By structuring problem solving into specialized roles and coordinating agent collaboration, MACI enables robust constraint awareness, self-verification, and adaptability, capacities absent in monolithic LLM architectures. The experimental results validate the effectiveness of MACI in improving planning consistency and satisfaction with constraints.

## 1. Introduction

Artificial intelligence requires capabilities beyond pattern matching. To tackle complex real-world tasks, AI must exhibit deliberate reasoning, temporal awareness, and effective constraint management. Although large language models (LLMs) (e.g. (OpenAI, 2024a; Anthropic, 2024; DeepSeek-AI et al., 2025)) excel at pattern recognition, they face significant challenges in planning tasks that require sustained attention, knowledge of constraints, and reasoning in both past and future temporal states (Kahneman, 2011).

### 1.1. Limitations of LLMs in Planning

LLMs reveal three limitations that fundamentally undermine their effectiveness in complex planning scenarios:

1. *Lack of Self-Verification*. LLMs struggle with validating their own output, a problem that extends beyond Gödel's incompleteness theorems for formal systems (Gödel, 1967). Their probabilistic nature and lack of logical foundations create significant barriers to self-assessment. This intrinsic limitation means LLMs cannot reliably detect errors or inconsistencies in their generated content (Hong et al., 2024; Weng et al., 2023; Stechly et al., 2024), necessitating external mechanisms to validate and refine their output.

2. *Attention Bias and Constraint Drift*. In complex scenarios, LLMs demonstrate a critical cognitive limitation known as cognitive tunneling. This phenomenon occurs when recently provided context dominates and progressively erodes earlier-established constraints (Wei et al., 2024; Momennejad et al., 2023). When planning a multi-leg journey, for instance, an LLM might optimize the final travel segment while completely neglecting crucial earlier constraints such as vehicle availability or required rest periods. This bias toward local optimization fundamentally undermines the global feasibility of generated plans.

3. *Lack of Common Sense Integration*. LLMs often overlook practical constraints that humans intuitively consider (Bhagavatula et al., 2020; McKenna, 2023). This deficiency becomes particularly evident in domains that require real-world experience and understanding. In travel planning, an LLM might generate a route without accounting for airport security processing times. In logistics, it may create schedules that ignore resource availability and preparation windows. Without explicit, granular specifications, these models produce plans that appear superficially coherent but remain impractical.

### 1.2. The MACI Framework

To address these limitations, we propose Multi-Agent Collaborative Intelligence (MACI), a framework designed to enhance reasoning and planning through a multi-component architecture. MACI introduces three core components:

1. *Meta-Planner (MP).* The meta-planner serves as the central orchestration mechanism in MACI. It analyzes task requirements, identifies roles and constraints, and dynamically generates a dependency graph (or workflow template) tailored to the task. This template includes actionable workflows with nodes representing roles (e.g., cook, driver, supervisor) and edges representing dependencies (e.g., temporal, spatial, or resource constraints). The incorporation of common sense augmentation into the metaplanner ensures that the generated plans are realistic, comprehensive, and aligned with practical constraints.

2. *Common and Task-specific Agents.* MACI employs two types of agents to execute the generated plans:

- *Common Agents*: These agents handle general-purpose tasks, including constraint validation, practical reasoning, and performance evaluation. For instance, a *Common Sense Integration* agent identifies implicit constraints, while a *Constraint Validation* agent ensures feasibility and compliance with the task's requirements.

- *Task-specific Agents*: These agents cater to domain-specific requirements, including task-dependent data and knowledge augmentation, selection of the most effective planning algorithms, safety and ethics assessment, and emergency response optimization. By integrating domain expertise, they extend the capabilities of common agents, enabling MACI to address specialized planning challenges with precision and adaptability.

3. *Run-Time Monitor.* The run-time monitor handles real-time adjustments to the static plan in response to unexpected changes, such as resource delays, environmental disruptions, or evolving task requirements. This component ensures adaptability and robustness by:

- Monitoring plan execution to detect deviations.

- Activating emergency agents to revise dependencies, reassign roles, or dynamically adjust constraints.

- Communicating updates to affected agents to maintain coherence throughout the workflow.

### 1.3. Summary: How MACI Addresses LLM Limitations

With its multi-component architecture, MACI directly addresses the three critical limitations of LLMs in planning:

1. *Lack of Self-Verification.* MACI separates planning from validation, employing independent agents for validation. These agents operate without shared memory or interference, ensuring external verification of outputs and mitigating the risks of self-referential errors.

2. *Attention Bias and Constraint Drift.* MACI avoids relying on a single LLM to execute complex, multi-step reasoning sequentially. Instead, it utilizes small collaborative agents that enjoy two key benefits: independence and

well-defined input/output protocols (ensuring specificity and quality) for specific tasks. These agents operate within restricted context windows of e.g., 1k tokens, which physically limits attention bias and ensures that earlier constraints are not overridden by recent context. By logically scoping problems and physically constraining context, MACI preserves global feasibility and mitigates cognitive tunneling.

3 *Lack of Common Sense Integration.* MACI incorporates a Common Sense Integration Agent and other specialized agents to identify implicit constraints and augment plans with practical, domain-specific knowledge. This ensures that generated plans are realistic, comprehensive, and aligned with real-world conditions.

Through its innovative architecture, MACI overcomes the inherent limitations of LLMs, enhancing their capacity for deliberate reasoning and planning. In subsequent sections, we demonstrate MACI's effectiveness through evaluations in complex scenarios, such as the Traveling Salesman Problem (TSP) and a multi-layered dinner planning task.

## 2. Related Work

The development of MACI builds on theoretical insights from formal systems and addresses limitations of current multi-agent architectures. Gödel's second incompleteness theorem (Kennedy, 2008; Gödel, 1967) established that no consistent formal system can prove its own consistency. This principle extends to LLMs, which rely on probabilistic rather than axiomatic foundations, making them inherently incapable of reliable self-validation. To address this, MACI employs a distributed validation architecture, where independent agents validate externally the output, bypassing the self-referential loops that may lead to inconsistencies.

In formal systems, consistency proofs require a "higher-order" system. Analogously, MACI provides a validation framework that operates as a higher-order metasystem for LLM output. By decoupling planning from validation, MACI mirrors the separation needed in formal systems, where validation is performed independently to avoid conflicts and errors.

Moreover, MACI advances the state of the art in multi-agent systems by addressing challenges that existing frameworks have not fully resolved.

Current multi-agent systems (MAS) primarily function as integration platforms for coordinating multiple LLMs. Notable frameworks include Microsoft's AutoGen (Wu et al., 2024), the Multi-LLM Agent Debate Framework (Du et al., 2023; Chang, 2023; 2024), LangGraph and CrewAI (LangChain AI, 2024; Moura, 2024), XAgent (XAgent Team, 2023), and CAMEL (Li et al., 2023). While these frameworks excel in agent coordination, they prioritize task

distribution over the comprehensive constraint management necessary for complex planning.

MACI bridges this gap by integrating a meta-planning module with independent agents that validate constraints, enabling robust and adaptable solutions in dynamic real-world scenarios. The meta-planner constructs task-specific dependency graphs that encode inter-agent constraints, ensuring precise role allocation while maintaining global feasibility. Meanwhile, validation agents, operating independently of the planning process, monitor for errors and inconsistencies stemming from probabilistic output, ensuring alignment with task objectives. This separation of roles mitigates cognitive tunneling and enhances adaptability, allowing MACI to dynamically respond to real-time disruptions such as resource shortages or environmental changes.

By integrating these advanced mechanisms, MACI goes beyond existing MAS frameworks to provide a cohesive architecture for complex reasoning and planning. It ensures a high degree of scalability and robustness, making it suitable for applications ranging from logistical optimization to adaptive decision-making in uncertain environments.

## 3. Case Study: Illuminating LLM Limitations

Planning methodologies fall into two categories: *sequential* and *reactive*. *Sequential planning* organizes time-ordered schedules (Allen & Hayes, 1989), anticipates future scenarios (Cox & Veloso, 1998), and improves through past experiences (Kolodner, 1993). *Reactive planning* adapts to dynamic conditions (Hammond, 1990), prioritizes immediate actions (Georgeff & Lansky, 1987), and leverages data-driven forecasting (Kushmerick et al., 1995).

This section examines LLM limitations in planning via two experiments: a scheduling problem exposing *sequential* planning challenges and a resource allocation task highlighting *reactive* planning deficiencies. Section 4 proposes solutions.

**Problem Statement**   We experiment with a Thanksgiving dinner planning task:

*Setup:*
- Mom (Sarah) hosts dinner at 6:00 PM in Boston.
- Family arrivals:
  - Dad (James) from San Francisco, lands at 1:00 PM ET.
  - Sister (Emily) from Chicago, lands at 2:30 PM.
  - Brother (Michael) driving from NY, arrives at 3:00 PM.
- Grandma, healthy, requires pickup from suburban Boston.

*Constraints:*

- James must rent a car post-landing.
- Emily needs an airport pickup (no alternatives).
- Turkey requires 4 hours to cook; someone must be home once it's in.
- Side dishes need 2 hours of preparation.
- Travel times:
  - Home to BOS Airport: 1 hour.
  - BOS Airport to Grandma's: 1 hour.
  - Home to Grandma's: 30 minutes.

*Key Planning Questions:*

1. When should cooking start?
2. Who picks up Emily, and when?
3. When and by whom should Grandma be picked up?

Using this problem, we examine sequential and reactive planning performed by GPT-4o and DeepSeek.

*Table 1.* DeepSeek Proposed Schedule

| Time | Task | Assigned |
|------|------|----------|
| 12:00pm | Sarah starts preparing side dishes. | Sarah |
| 1:00pm | James arrives at the airport, begins car rental (takes 30-60 minutes). | James |
| 2:00pm | James is ready to pick up Emily; Sarah and James put the turkey in the oven. | J, S |
| 2:30pm | Emily lands at BOS. | Emily |
| 3:00pm X | James arrives at airport to pick up Emily | J |
| 3:00pm | Michael arrives at home and leaves to pick up Grandma. | Michael |
| 3:15pm X | Michael at Grandma's home. | M |
| 3:45pm | Grandma arrives home; Michael returns. | M |
| 4:00pm | James and Emily arrive home. | J, E |
| 6:00pm | Dinner is served. | All |

### 3.1. Study #1 Sequential Planning

GPT-4o and DeepSeek struggled with real-world travel logistics, omitting key human considerations such as luggage claim time, rental car processes, and spatial relationships between airport terminals and the rental center (Table 1). These gaps forced manual constraint additions, highlighting LLM's inability to integrate experiential knowledge, a gap addressed by the MACI common sense agent.

DeepSeek's schedule further revealed spatial-temporal errors: 1) Spatial: Assumed James drove home immediately after renting a car at Boston Logan, ignoring his airport location while awaiting Emily; and 2) Temporal: Directed Michael to return home before heading to Grandma's, missing the optimal direct route from NYC.

Table 2 shows the GPT-4o schedule, which appears feasible but contains two critical errors in the case of adaptive planning required for emergency: 1) Arithmetic: Incorrectly calculates Grandma's round-trip driving time as 30 minutes (vs. 30×2 minutes); and 2) Over-Constraint: Assumes only Sarah must watch the oven (vs. "someone"), creating brittleness under reduced slack time (e.g., delays).

*Table 2.* GPT4o Proposed Schedule

| Time | Task | Assigned |
|---|---|---|
| 1:00pm | James lands in Boston | James |
| 2:00pm | Turkey goes into the oven | Sarah |
| 2:00pm | James finishes car rental | J |
| 2:30pm | Emily lands at BOS | Emily |
| 2:30pm | James picks up Emily at airport | J |
| 3:00pm | Michael arrives home | Michael |
| 4:00pm | Side dishes preparation starts | S, M |
| 5:00pm | Michael leaves to pick up Grandma | M |
| 5:30pm X | Michael arrives home with Grandma | M |
| 6:00pm | Dinner is served | All |

Analysis (with detailed execution in Appendix A) links both errors to flawed reasoning in constraint interpretation.

**Diagnoses: Common-Sense Constraints and Isolated Processing Syndrome**   LLMs struggle with implicit real-world constraints that humans consider common sense, limiting their planning capabilities. Additionally, we identify *isolated processing syndrome*, where LLMs tackle sub-tasks independently, lacking awareness of overall constraints. This results in two critical failures: missing obvious optimizations or generating infeasible plans by violating stated constraints.

### 3.2. Study #2 Reactive Planning

Real-world scenarios do not always follow plans precisely. Robust systems require contingency planning for factors such as weather, traffic, or airline changes. These cascade through schedules, demanding adaptive replanning.

To evaluate LLMs' dynamic replanning, we introduce a major disruption in our Thanksgiving scenario: James's flight is delayed by 3 hours (arrival 4:00 PM vs. 1:00 PM). This forces adjustments to pickups, meal prep, and coordination while preserving original constraints.

LLM responses reveal critical flaws: 1) DeepSeek violates core constraints by unjustifiably delaying dinner to 7:00 PM (vs. the 6:00 PM deadline); and 2) GPT-4o (Table 5 in **Appendix** D) commits a safety violation: leaving the oven unattended, despite explicit constraints. These errors highlight LLMs' inability to reliably maintain and validate constraints during replanning, even with full information.

**Diagnosis: Attention Narrowing**   Claude detects constraint violations in other LLMs' plans, but GPT-4o and DeepSeek struggle with self-validation, revealing an asymmetry in error detection. LLMs often misinterpret constraints during planning (e.g., rigidly enforcing "someone must be in the house" while the turkey is in the oven), propagating errors throughout their reasoning.

Two key limitations emerge: 1) *Attention narrowing*: Overfocusing on objectives (e.g., arrival times) leads to neglect of critical constraints (e.g., fire safety). 2) *Solution rigidity*: Once a constraint is satisfied (e.g., assigning Sarah to oven

duty), LLMs treat it as fixed, failing to explore alternatives.

For example, GPT-4o assigned Sarah to monitor the oven but failed to reallocate this task to Grandma earlier, missing an efficiency gain by freeing Sarah to serve as an additional driver.

### 3.3. Summary of LLM Limitations in Planning

Our analysis reveals three core limitations in LLM-based reasoning methods (CoT (Wei et al., 2022), ToT (Yao et al., 2023)):

**Metacognitive Limitations**   LLMs struggle with self-validation and constraint awareness. While external models detect errors in others' plans, planners overlook their own (e.g., GPT-4o rigidly assigning Sarah to oven duty without considering Grandma's availability). Causes include:

1. Pattern-matching over analytical validation

2. Lack of belief-state tracking

3. Single-solution focus vs comparative reasoning

Current reasoning methods reinforce these flaws by operating within the same cognitive framework.

**Attention Bias**   Transformers prioritize recent context, leading to: 1) *Narrowing*—recent constraints (e.g., arrival times) overshadow earlier ones (e.g., oven safety); and 2) *Isolated Processing*—sub-tasks are handled independently without global awareness.

**Common Sense Gaps**   LLMs fail to infer implicit real-world knowledge (e.g., luggage claim times, rental logistics), requiring explicit specification of human-obvious constraints (e.g., airport-terminal proximity).

In Sections 4 and 5, we show how MACI's meta-planner corrects errors and dynamically adapts plans.

## 4. MACI Framework Specification

MACI implements a three-component architecture to address current LLM limitations: metacognitive constraints, attention bias, and gaps in common-sense reasoning. Each component plays a distinct role in enabling robust and adaptable planning capabilities.

### 4.1. Three-Component Architecture

**Meta-Planner Component**   The meta-planner $\mathcal{MP}$ functions as a higher-order planner that generates task-specific planning systems:

$$\mathcal{MP} : (\mathcal{O}, \mathcal{C}_{\mathcal{E}}) \rightarrow \mathbf{W},$$

where $\mathbf{W}$ represents a planning system composed of specialized, coordinated agents. Similar to a compiler generator producing compilers from specifications, $\mathcal{MP}$ constructs

agent networks from task requirements. It analyzes objectives, identifies required roles and dependencies, selects appropriate agents, and establishes interaction protocols. This produces a workflow template that defines the planning state space and the coordination mechanisms needed to solve the task.

**Agent Repository Component** This component maintains a distributed collection of planning agents, each designed with a restricted context window and specialized interface. By dividing cognitive tasks among agents, the repository ensures a complete representation of constraints without overwhelming individual components. The meta-planner queries this repository to select agents for specific roles and dependencies based on task requirements.

**System Infrastructure Component** Built on open-source multi-agent system (MAS) frameworks, the infrastructure component supports essential operations such as agent registration, message routing, resource allocation, and deployment scaling. This foundation provides the necessary runtime environment for executing and monitoring the generated workflows.

### 4.2. Agent Repository Design

The agent repository in MACI serves as a structured database, enabling efficient registration, retrieval, and matching of agents to task requirements. By categorizing agents into *common agents* and *specialized agents*, the repository supports both generalized functionality and domain-specific expertise, as outlined in Section 4.1.

#### 4.2.1. LIGHTWEIGHT, INDEPENDENT AGENT DESIGN

MACI avoids relying on a single LLM to execute complex, multi-step reasoning sequentially. Instead, it utilizes *small, independent agents* that adhere to strict efficiency and modularity principles. These agents operate with well-defined input/output protocols and are constrained to *restricted context windows* to mitigate attention bias and prevent earlier constraints from being overridden by recent context.

By *scoping problems logically* and *constraining context physically*, MACI ensures that each agent processes only the task-relevant information needed for its specific role. This design prevents cognitive tunneling, maintains global feasibility, and enhances robustness in dynamic environments.

#### 4.2.2. AGENT REGISTRATION AND SPECIFICATIONS

Each agent is registered in the repository using a standardized protocol buffer that encodes the following attributes:

- *Input/output protocol ($P$)*: Defines the data format and expected interactions for seamless communication.

- *Agent type ($t$)*: Specifies whether the agent is *common* or *specialized*.

- *Capability vector ($c$)*: Encodes the agent's functional capabilities, constraints, and operating conditions.

- *Context window size ($w$)*: Ensures that each agent operates within a restricted buffer ($w \leq 1k$ tokens) to prevent attention bias and excessive information retention.

- *Computational efficiency constraint ($e$)*: Agents are lightweight, avoiding unnecessary memory usage or processing delays.

- *User rating ($r$)*: Tracks historical performance evaluations to prioritize reliable agents during selection.

The meta-planner retrieves agents from the repository using a three-step matching process:

1. *Task-to-capability matching*: Filters agents based on their capability vector ($c$) and task-specific requirements.

2. *Protocol verification*: Ensures compatibility of input/output protocols ($P$) between selected agents to prevent communication errors.

3. *Agent ranking*: Ranks agents by their relevance, efficiency, and historical user rating ($r$) to select the optimal candidates.

This structured retrieval mechanism ensures that MACI efficiently scales to complex planning problems without requiring predefined agent hierarchies. By leveraging protocol buffers and a structured repository, MACI achieves both modularity and adaptability, allowing new agents to be introduced seamlessly while maintaining coherence across multi-agent interactions.

#### 4.2.3. STATE SPACE AND AGENT DESIGN

Tasks in MACI are modeled in a general five-dimensional state space to ensure comprehensive representation of constraints and dependencies. These dimensions include:

1. *Who (Actors)*: Identifies roles, constraints, and transitions between agents or individuals.

2. *Where (Location)*: Tracks physical or logical positions, transitions, and access rules.

3. *When (Time)*: Captures temporal constraints such as deadlines, durations, and time points.

4. *What (Resources)*: Manages resource availability, constraints, and associated costs.

5. *Why (Logic)*: Encodes rationale, dependencies, and risk assessments for decision-making.

This structured state space allows the meta-planner to generate workflows that account for all relevant constraints and dependencies across diverse domains.

### 4.2.4. AGENT ROLES IN STATE SPACE MANAGEMENT

**Common Agents**   Common agents are designed to handle foundational planning tasks that align with MACI's state space dimensions (*Who, Where, When, What, Why*). These agents provide general-purpose functionality that ensures consistency, feasibility, and robustness across planning tasks. Their primary responsibilities include:

- *Constraint Validation Agents*: Ensure adherence to temporal, spatial, and resource constraints by verifying the feasibility of generated plans.
- *Common Sense Integration Agents*: Identify implicit constraints that may be overlooked, such as transition times, dependencies, or practical limitations.
- *Adaptation Agents*: Dynamically adjust plans in response to changes in task environments, such as resource delays or evolving requirements.
- *Performance Evaluation Agents*: Assess the quality and efficiency of proposed plans relative to predefined metrics, ensuring continuous improvement.

By addressing these tasks, common agents form the backbone of MACI's planning architecture. Their modular design enables reuse across multiple domains, and their collaborative functionality ensures they work seamlessly with specialized agents to maintain global consistency and coherence within the planning workflow.

**Task-Specific Agents**   These agents cater to domain-specific requirements, including task-dependent data and knowledge augmentation, selecting and optimizing planning algorithms, safety and ethics assessment, and emergency response optimization. By leveraging domain expertise, specialized agents extend the capabilities of common agents, enabling MACI to address specialized planning challenges with precision and adaptability.

### 4.2.5. SEAMLESS INTEGRATION AND SCALABILITY

The repository's standardized agent specifications and matching mechanism enable MACI to scale efficiently across domains. By leveraging modular designs and protocol buffers, new agents can be integrated seamlessly into existing workflows, ensuring adaptability and extensibility without compromising performance or consistency.

### 4.3. Meta-Planner: Planning a Planner to Plan

The mission of the meta-planner $\mathcal{MP}$ is to construct a planner that generates an actionable workflow for a given task. It does so by analyzing task objectives, identifying roles and constraints, and organizing agents into a structured execution plan. This three-phase approach ensures that every agent and dependency is optimally placed, refined, and validated, leading to robust, task-specific workflows.

### 4.3.1. THE META-PLANNER ALGORITHM

**Appendix** E provides the full algorithm.

The meta-planner operates as a higher-order planning system that formulates workflows as directed graphs:

$$\mathbf{W} = (\mathcal{N}, \mathcal{E}), \text{ where } \mathcal{N} = A_n^*, \mathcal{E} = A_e^*. \quad (1)$$

Here, $\mathcal{N}$ denotes roles assigned to agents, and $\mathcal{E}$ represents dependencies between roles, including constraints such as timing, data flow, and supervision requirements.

### 4.3.2. META-PLANNING DESIGN ELEMENTS

**Role and Qualification Analysis**   The meta-planner extracts roles from task objectives and maps them to required qualifications:

$$\text{map}_{\text{role}} : \mathcal{O} \to \{(n_i, q_i)\} \quad (2)$$

where $n_i$ represents a role and $q_i$ its required qualifications (e.g., a driver requires a license, a cook requires experience).

**Constraint Management**   Constraints govern role interactions and dependencies. The framework maintains a global constraint set:

$$C = C_E \cup C_I \cup C_D \quad (3)$$

where $C_E$ represents explicit constraints from task specifications, $C_I$ denotes implicit constraints identified by common sense agents, and $C_D$ represents derived constraints from agent interactions.

**Agent Assignment**   Two categories of agents are assigned based on task requirements:

- *Node Agents (Role Execution)*:

$$A_n^* = \underset{A_i \in \mathbf{A}}{\arg\min} \sum_{n_j} \text{dist}(q_j, A_i.\text{capabilities}) \quad (4)$$

  These agents are responsible for fulfilling role qualifications and managing people-role assignments.

- *Edge Agents (Dependency Management)*:

$$A_e^* = \underset{A_i \in \mathbf{A}}{\arg\min} \sum_{e_j} \text{dist}(c_j, A_i.\text{capabilities}) \quad (5)$$

  These agents ensure dependencies between roles are correctly maintained, such as time constraints, spatial relations, and supervisory requirements.

### 4.4. Workflow Execution Framework

The final workflow $\mathbf{W}^*$ must be executed in a runtime environment. In this work, we evaluate $\mathbf{W}^*$ by entering it into an LLM (e.g., GPT4o) alongside the problem statement. A key limitation is that the *feedback loop for refining* $\mathbf{W}^*$ *is currently manual*, requiring iterative adjustments to optimize execution. Future research will focus on automating this process to enhance adaptability and efficiency.

# 5. Evaluating $\mathcal{MP}$ vs. Independent LLMs

To assess $\mathcal{MP}$'s performance and adaptability, we adopted a dual-approach experimental structure. The first experiment uses the Traveling Salesperson Problem (TSP) to validate $\mathcal{MP}$'s optimization capabilities. The second involves the Thanksgiving Dinner Planning problem, showcasing $\mathcal{MP}$'s ability to handle complex, real-world challenges with cross-thread dependencies and dynamic adaptability. Due to space constraints, detailed results for these experiments are provided in **Appendices F** and **G**, respectively.

## 5.1. Traveling Salesperson Problem (TSP)

The TSP experiment benchmarks $\mathcal{MP}$ against standalone planners (Claude (Anthropic, 2024), DeepSeek R1 (DeepSeek-AI et al., 2025), GPT-4o (OpenAI, 2024a)) and their $\mathcal{MP}$-integrated counterparts. The metrics include solution quality and optimality.

**Result Summary**   Without $\mathcal{MP}$, DeepSeek performs best, while Claude and GPT-4o struggle, each exceeding the optimal travel time by more than $10\%$. With $\mathcal{MP}$, Claude requires two iterations to reach the optimal distance, while both GPT-4o and DeepSeek solve the problem in a single attempt.

Although TSP involves a straightforward single-thread planning process, $\mathcal{MP}$ still provides notable improvements. Again, see **Appendix F** for details.

## 5.2. Thanksgiving Dinner Planning

This task, detailed in Section 3, evaluates $\mathcal{MP}$'s ability to generate workflows $\mathbf{W}^*$ with enhanced constraint and dependency management in the MACI setting. Unlike TSP, this problem involves multiple interdependent agents, introducing complex coordination challenges.

Planning performance is assessed across three configurations: DeepSeek + $\mathcal{MP}$, GPT-4o + $\mathcal{MP}$, and Claude + $\mathcal{MP}$. The prior results in Section 3 show that all LLMs fail the task when executed independently.

Evaluation metrics include:

$$\text{Performance} = \{\%\text{Constraint satisfaction, Flexibility}\},$$

where flexibility measures slack time incorporated to handle unexpected events.

### 5.2.1. META-PLANNING FOR THANKSGIVING EVENT

Following **Algorithm 1**, $\mathcal{MP}$ generates workflows with:

- Role nodes (e.g., cook, drivers, supervisor),
- Explicit constraint edges (e.g., temporal, spatial, etc.),
- Implicit constraint edges from common-sense analysis.

The planner monitors nodes and edges, enabling dynamic adjustments. The full specifications are in **Appendix G**.

**Evaluation Scenarios**   We test $\mathcal{MP}$ under:

1. *Sequential Planning*: Task executed as planned.

2. *Reactive Planning*: A 3-hour flight delay requiring task reallocations.

**Meta-Planner Output**   $\mathcal{MP}$ enhances planning by:

- Identifying implicit constraints (e.g., luggage claim time, car rental delays),
- Clarifying role dependencies,
- Incorporating common-sense constraints (e.g., fatigue, social preferences),

In reactive planning, $\mathcal{MP}$ integrates an *alert agent* to detect flight delays at departure, enabling timely workflow updates and demonstrating adaptability.

### 5.2.2. EXPERIMENTAL RESULTS

**Sequential Planning Performance**   With $\mathcal{MP}$'s enhanced workflow $\mathbf{W}^*$, all three LLMs successfully generated feasible solutions, a significant improvement over their previous failures with the original problem specification.

*Table 3.* Sequential Planning Performance. (# = iterations)

| LLM | # | Notable Features |
|---|---|---|
| **DeepSeek** | 2 | Optimized airport wait time for James; balanced workload |
| GPT4o | 3 | Extra travel for Michael; suboptimal load balance |
| Claude | 2 | Unnecessary travel between pickup tasks |

*Table 4.* Reactive Planning Performance (Alert: flight delay)

| LLM | # | Notable Features |
|---|---|---|
| **DeepSeek** | 3 | Smart routing of Michael directly to airport; efficient travel patterns |
| GPT4o | **X** | Failed to maintain critical constraints; unable to recover |
| Claude | 3 | Two valid plans with different trade-offs; longer wait times |

Table 3 summarizes the detailed schedules documented in Tables 13, 14, and 15, in **Appendix G.8**. DeepSeek demonstrated superior scheduling efficiency by optimizing James's airport wait time for Emily's pickup, requiring only two iterations. While GPT4o eventually produced a valid solution in three iterations, it created suboptimal travel patterns by having Michael make separate trips. Claude's solution, though feasible in two iterations, included unnecessary travel between pickup tasks. This experiment highlighted how $\mathcal{MP}$'s explicit constraint specification and common-sense augmentation enabled consistent performance improvement across different LLMs.

**Reactive Planning Performance** The flight delay scenario revealed significant differences between LLMs' capabilities. DeepSeek demonstrated superior spatial reasoning by routing Michael directly to the airport, an insight that should have come from $\mathcal{MP}$'s common-sense spatial reasoning. This unexpected ability to improve workflow highlights the synergy between $\mathcal{MP}$ and LLM —$\mathcal{MP}$ provided early alert through its information agent (Table 16 in **Appendix** G.9).

Table 4 summarizes the detailed schedules documented in Tables 17, 19, and 20, in **Appendix** G.9. DeepSeek leveraged the early alert at 10:00 AM for immediate replanning. In contrast, Claude produced two feasible plans but missed the 10:00 AM alert in $\mathbf{W^R}$, starting its schedule at 1:00 PM and missing opportunities for proactive actions like early Grandma pickup to free Sarah as a driver. GPT4o failed entirely, producing three constraint violations it could not recognize, preventing further improvements.

# 6. Alternate Views and Conclusion

The limitations of current LLMs in planning and reasoning are well documented: reliance on pattern matching over deliberate reasoning, inability to self-validate, and failure to maintain constraints, compounded by well-known hallucinations and biases. These are not incidental flaws, but rather **structural weaknesses inherent to LLM architectures**. Addressing these shortcomings requires more than incremental improvements; it demands a fundamental rethinking of AI's approach to planning, constraint management, and validation.

## 6.1. MACI as the Necessary Evolution of AI Planning

MACI represents this necessary evolution. Its **structured meta-planning, distributed validation, and proactive multi-agent coordination** systematically overcome the deficiencies of single-LLM architectures. Empirical evaluations substantiate its effectiveness:

- In the **Thanksgiving Dinner Planning experiment**, MACI successfully resolved **intricate cross-thread dependencies** that individual LLMs failed to handle.

- Its ability to **dynamically reallocate tasks, integrate implicit constraints, and adapt to evolving conditions** underscores MACI's real-world applicability.

These results, along with the **Traveling Salesperson Problem (TSP) experiment**, demonstrate a **fundamental shift in AI reasoning and planning**, rather than mere marginal improvements. While there is still room for enhancement, approaches such as self-refinement using GRPO (Hong et al., 2024; OpenAI, 2024b), supervised fine-tuning (Anthropic, 2024), distillation (Zhang & colleagues, 2023), and information theory-based techniques could further advance AI towards AGI.

## 6.2. Alternative Approaches and Their Shortcomings

**A. Single-LLM Enhancements: The "Average Model Problem"** Efforts to enhance single LLMs, by adding memory modules, logical reasoning layers, or constraint-aware training, ultimately face the **average model problem**: any single model trained to handle diverse planning domains must make trade-offs, reducing its effectiveness in specific tasks. As seen in (Weng et al., 2023), single LLMs trained in various planning domains struggle to maintain peak performance in all scenarios, requiring trade-offs that degrade specialized reasoning capabilities.

**Single LLMs also struggle to address the challenge of self-verification**, as an LLM remains bound by its probabilistic reasoning and cannot independently validate its own plans, a limitation that extends beyond Gödel's incompleteness theorems for formal systems (Gödel, 1967). Their lack of logical foundations and reliance on probabilistic inference create significant barriers to self-assessment, necessitating external validation mechanisms (Hong et al., 2024; Weng et al., 2023; Stechly et al., 2024).

**B. Multi-Agent Systems (MAS): Limited Global Coordination and Static Configuration** MAS facilitate distributed problem-solving but lack global coordination and real-time adaptability. Agents optimize locally with limited information sharing, often leading to suboptimal system-wide outcomes (Stone & Veloso, 2000; Wooldridge, 2009).

While MAS can incorporate agents like RAG for data augmentation or CoT for abductive reasoning, these additions are predefined rather than dynamically integrated via feedback, limiting responsiveness to changing environments. The inability to reconfigure in real time constrains MAS in complex, evolving scenarios. Unlike MAS, MACI dynamically adjusts task allocation based on real-time feedback, as demonstrated in the Thanksgiving dinner experiment (e.g., responding to flight delays).

**C. Additional Views** AI planning and reasoning are vast fields with ongoing research exploring various alternative approaches. Additional details are provided in **Appendix** A.

## 6.3. Final Statement

When designed correctly, multi-LLM agent collaboration can mitigate hallucinations and biases while significantly enhancing reasoning and planning, as demonstrated in this paper. The future of AI planning does not lie in incremental improvements to LLMs but in redefining the very structure of intelligence. MACI embodies this paradigm shift, offering a **scalable, adaptable, and verifiable** framework for AI-driven reasoning and decision-making. **MACI is the blueprint for this transformation**.

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

# Appendices

## A. Advancing Multi-Agent Systems: The Next Evolution with MACI

Multi-Agent Collaborative Intelligence (MACI) represents the next stage in the evolution of multi-agent systems, advancing beyond traditional frameworks to enable more dynamic, adaptable, and self-improving AI interactions. Current Large Language Models (LLMs) operate at system-1 level, excelling in pattern recognition and linguistic generation, but lacking high-level **reasoning and planning** capabilities, key hallmarks of system-2 intelligence (Kahneman, 2011; Bommasani et al., 2021). MACI aims to bridge this gap by enabling structured multi-agent collaboration to enhance decision-making, adaptability, and self-improvement. Beyond previous discussions, three key areas require further development: (1) conducting more experiments and establishing benchmarks to validate progress, (2) improving reasoning quality, and (3) developing a feedback loop for self-improvement while managing domain-specific constraints.

**Key Areas for MACI Enhancement**

**Benchmarking and Empirical Validation** For MACI to advance, rigorous empirical validation is essential. Although theoretical improvements provide direction, practical validation ensures robustness. Establishing benchmarks will enable structured evaluations in multiple domains, offering measurable comparisons to assess reasoning quality, adaptability, and real-time decision-making efficiency (Hendrycks et al., 2021).

**Enhancing Reasoning and Planning Capabilities** A critical distinction between MACI and single-LLM architectures is the emphasis on structured reasoning and strategic planning. Unlike traditional multi-agent systems that rely on heuristic-based coordination, MACI should incorporate dynamic logic refinement, probabilistic inference mechanisms, and hierarchical task planning (Georgeff et al., 1998; Yao et al., 2023). These enhancements enable MACI to go beyond shallow pattern recognition and improve complex decision making in real-world applications.

**Feedback-Driven Self-Improvement** For MACI to be truly adaptable, it must incorporate a feedback loop that enables self-improvement. However, not all errors can or should be corrected at the system level. Some errors arise due to individual circumstances, such as determining seating arrangements at a banquet table. Constraints related to tasks, domains, and cultures are highly contextual and difficult to model comprehensively. Instead of modifying MACI itself, such context-dependent constraints should be handled via context augmentation techniques, such as Retrieval-Augmented Generation (RAG) (Lewis et al., 2020) and few-shot prompting (Brown et al., 2020), to dynamically adapt solutions based on user preferences and contextual inputs.

**Balancing Systemic and Task-Specific Improvements** The distinction between system-wide enhancements and task-specific adjustments is crucial. Core agents within MACI should undergo continuous improvements in reasoning, coordination, and adaptability. However, user-specific requirements—such as subjective preferences in planning—should remain flexible and be addressed at the application layer. This division ensures that MACI remains both robust in its general reasoning capabilities and adaptable to varying user needs without overcomplicating its core architecture.

**Alternative Approaches and Future Directions** A key question for MACI is whether Reinforcement Learning (RL) is necessary for personalization, improvement, or alignment with values (e.g., national, corporate). Although RL has been effective in optimizing reward-driven behaviors, its applicability to personalize MACI remains questionable due to challenges such as data sparsity (Ouyang et al., 2022), conflicting preferences (Christiano et al., 2017), and unstable reward signals (Ziegler et al., 2019).

Instead of relying on RL-based adaptation, alternative methods should be explored based on specific applications. Retrieval-Augmented Generation (RAG) can fetch relevant personal data without modifying the model (Lewis et al., 2020), while meta-prompting allows for dynamic preference injection (Reynolds & McDonell, 2021). Graph-based optimization can manage constraints more effectively than reinforcement learning, and rule-based filtering can enable value alignment without extensive retraining.

**Considerations for MACI: When to Improve and When to Avoid Adaptation** MACI should focus on logical consistency, constraint satisfaction, and dynamic adaptation while avoiding excessive personalization that leads to overfitting. Some constraints, such as cultural or domain-specific preferences, should be handled dynamically rather than embedded into MACI's core framework. Using structured, context-sensitive, and retrieval-based adaptation mechanisms, MACI can ensure both flexibility and robustness.

**Conclusion** MACI advances multi-agent systems by improving **reasoning, planning, and self-improvement**, distinguishing itself from system-1-level LLMs that rely solely on pattern recognition. By addressing the fundamental limitations of current AI architectures, MACI enables structured decision-making and real-time adaptability. Although

personalization remains a challenge, domain-specific constraints should be managed dynamically instead of hard-coded into the MACI architecture. Future research should focus on structured benchmarking, logic refinement, and context-aware adaptation mechanisms to further the evolution of MACI.

## B. Validation and Recovery Protocols

The validation protocol implements a multi-stage process for ensuring state consistency. When any agent proposes a state change, the validation agent initiates a sequence of checks:

$$\text{validate}(s_t \to s_{t+1}) = \begin{cases} \text{true} & \text{if all checks pass} \\ \text{false} & \text{if any check fails} \end{cases} \quad (6)$$

The protocol begins with pre-validation. Before a state transition starts, the validation agent queries relevant agents about preconditions. For a travel booking, temporal agent verifies the proposed times fit within existing schedules. Spatial agent confirms the physical feasibility of movements between locations. Role agent checks if all actors can perform their assigned functions.

During the transition, the protocol maintains atomic operations. The validation agent tracks changes across all state dimensions, ensuring partial updates cannot create inconsistent states. If the temporal agent approves a flight time but the resource agent finds insufficient seats, the entire transition fails and rolls back.

Post-validation examines the resulting state. The validation agent verifies that all constraints remain satisfied after the change. Common sense agent reviews the new state for practical issues that formal checks might miss. Strategy agent confirms the transition aligns with overall planning objectives.

When validation fails, the protocol triggers a structured recovery process:

$$\text{recover}(s_t, s_{\text{failed}}) \to s_{\text{valid}} \quad (7)$$

Recovery begins by logging the failure cause and violated constraints. The strategy agent then works with domain agents to generate alternative proposals that satisfy the constraints. This might involve relaxing non-critical constraints or exploring different approaches to meet the planning objectives.

### B.1. Operations Research Techniques in Validation Protocols

The validation protocols described above align closely with established methods in operations research (OR). Some relevant techniques include:

- **Constraint Programming (CP)**: Focuses on solving combinatorial problems by enforcing constraints, ensuring consistency across dimensions such as temporal, spatial, and resource availability (Rossi et al., 2006).

- **Mixed-Integer Linear Programming (MILP)**: Optimizes decision variables subject to linear constraints and objective functions, often used in scheduling and resource allocation (Wolsey & Nemhauser, 1998).

- **Network Flow Algorithms**: Validates feasibility and optimizes flows in networks by ensuring capacity, timing, and availability constraints are satisfied (Ahuja et al., 1993).

- **Dynamic Programming (DP)**: Decomposes problems into sequential subproblems, useful for validating processes like inter-terminal walking or luggage claiming time (Bellman, 1957).

- **Monte Carlo Simulation**: Simulates scenarios to validate feasibility and robustness under uncertainty (Metropolis & Ulam, 1949).

- **Robust Optimization**: Focuses on solutions that remain feasible under uncertainty, ensuring plans adapt to disruptions (Ben-Tal & Nemirovski, 2009).

**Integration in Multi-Agent Systems** The validation agent also employs techniques from multi-agent systems, such as:

- **Blackboard Systems**: A shared workspace for collaborative validation by different agents, ensuring global consistency (England & Engelmore, 1987).

- **Consensus Protocols**: Used for distributed validation, where agents negotiate to ensure all constraints are met (Ren & Beard, 2005).

By combining these OR techniques with agent-based systems, the validation protocol ensures comprehensive and adaptive checks for workflow consistency. Future work can explore integrating heuristic methods, such as genetic algorithms or simulated annealing, to further enhance recovery processes.

## C. MACI **Additional Design Considerations**

### C.1. Cross-Domain Generalization

While the state space dimensions—Who, Where, When, What, and Why—are broad enough to cover diverse domains, additional customization may be required for unique applications. This section examines how MACI generalizes across domains like financial planning, healthcare logistics,

and supply chain optimization. The travel planning example is illustrative, emphasizing how MACI dynamically adapts state spaces and agents to domain-specific requirements.

### C.2. Dynamic Agent Registration and Evolution

This section explores how agents are dynamically developed, trained, and validated for new tasks. It discusses mechanisms for evaluating new agents and integrating them into the repository without retraining the entire system, ensuring scalability and adaptability.

### C.3. Scalability and Resource Efficiency

As the number of agents and task complexity grows, MACI employs strategies to manage communication overhead and optimize agent interactions. This section details techniques for clustering agents and hierarchical coordination to maintain scalability.

### C.4. Empirical Evaluation Across Domains

To demonstrate MACI's adaptability, this section presents empirical results from applying the framework to multiple domains. Examples include financial portfolio management, urban traffic planning, and hospital resource allocation, highlighting MACI's advantages over state-of-the-art systems.

### C.5. Challenges and Future Directions

While MACI addresses many limitations of LLM-based planning, challenges remain in real-time coordination, implicit knowledge integration, and robust recovery mechanisms. This section proposes future research directions to enhance MACI's performance and applicability to novel tasks.

## D. Additional Tables and Figures

### D.1. Case Study Tables

*Table 5.* GPT4o Revised Thanksgiving Schedule. Hazard! No one home watch oven between 3:00pm and 4:00pm.

| Time | Task | Assigned |
|---|---|---|
| 2:00pm | Turkey placed in oven (4-hour cooking time begins) | Sarah |
| 3:00pm | Michael arrives home | Michael |
| | Michael departs to pick up Emily from airport | Michael |
| 3:00pm X | Sarah departs to pick up Grandma | Sarah |
| 3:30pm | Arrive at Grandma's house | Sarah |
| 4:00pm | Arrive at airport for Emily's pickup | Michael |
| | Sarah home with Grandma | - |
| | James's flight lands | James |
| | Begin side dish preparation | Sarah |
| 4:30pm | James completes car rental process | James |
| 5:00pm | Michael returns home with Emily | - |
| 5:30pm | James arrive home | - |
| 6:00pm | Thanksgiving dinner served | Everyone |

### D.2. Notation

Table 6 presents all symbols used throughout this paper.

*Table 6.* Symbol Definitions

| Symbol | Definition | Symbol | Definition |
|---|---|---|---|
| *Basic Sets* | | | |
| $\mathcal{O}$ | Planning objectives | $\mathcal{P}$ | Available people |
| $\mathcal{C}_{\mathcal{E}}$ | Explicit constraints | $\mathcal{C}_{\mathcal{I}}$ | Implicit constraints |
| $\mathcal{M}$ | Performance metrics | $\mathcal{Q}$ | Role qualifications |
| *Workflow Components* | | | |
| $\mathbf{W}$ | Workflow network | $\mathcal{N}$ | Roles (nodes) |
| $\mathcal{E}$ | Dependencies (edges) | $\mathbf{A}$ | Agent repository |
| $n_i$ | Individual role | $e_{ij}$ | Role dependency |
| *Agent Functions* | | | |
| $f_{\text{role}}$ | Role-agent mapping | $f_{\text{edge}}$ | Edge-agent mapping |
| $V(\cdot)$ | Validation function | $\text{dist}(\cdot)$ | Capability distance |
| $\text{map}_{\text{role}}$ | Role extraction | $\text{map}_{\text{edge}}$ | Dependency extraction |
| *Optimization* | | | |
| $A_n^*$ | Selected node agents | $A_e^*$ | Selected edge agents |
| $\mathbf{W}^*$ | Optimal workflow | $V^*$ | Best validation score |

### D.3. State Space Dimensions and Description

*Table 7.* State Space Dimensions and Components

| Dimension | Core Components | Example States |
|---|---|---|
| Who (Actors) | • Actor ID
• Current Roles
• Role Constraints
• Role Changes | • Driver vs Passenger
• Supervisor vs Worker
• Buyer vs Seller
• Multiple role conflicts |
| Where (Location) | • Current Position
• Target Location
• Transition Points
• Access Rules | • Physical locations
• Virtual positions
• State transitions
• Boundary constraints |
| When (Time) | • Time Points
• Duration
• Deadlines
• Probability | • Event timestamps
• Process duration
• Completion times
• Delay likelihood |
| What (Resources) | • Methods
• Requirements
• Constraints
• Costs | • Tools/Vehicles
• Tickets/Permits
• Capacity limits
• Time-money trade-offs |
| Why (Logic) | • Rationale
• Dependencies
• Risks
• Alternatives | • Decision basis
• Causal chains
• Failure modes
• Backup plans |

### D.4. Example Common Agents

1. *Role Manager Agent (Who)*: Tracks actors, their roles, and their associated constraints, ensuring that all role-based requirements are satisfied.

2. *Spatial Agent (Where)*: Manages geographic and location-based constraints, verifying transitions between physical or virtual locations.

3. *Temporal Agent (When)*: Handles scheduling, timing, and deadlines, ensuring alignment with temporal constraints.

4. *Resource Agent (What)*: Tracks real-world resources such as tools, vehicles, or financial instruments, managing capacity, availability, and associated costs.

5. *Reasoning and Explanation Agent (Why)*: Maintains the rationale behind decisions, dependencies, and alternative plans, enabling consistent alignment with objectives and providing explanations for outcomes.

6. *Common Sense Agent*: Identifies implicit constraints, integrates practical knowledge, and ensures plans align with real-world considerations.

7. *Constraint Validation Agent*: Ensures that all constraints are satisfied and that proposed plans remain feasible.

8. *Plan Evaluation Agent*: Assesses the effectiveness of plans against predefined metrics and objectives.

9. *What-If Testing Agent*: Evaluates plan robustness by simulating alternative scenarios and analyzing their impact.

10. *Compliance and Safety Agent*: Monitors adherence to safety standards, ethical principles, and regulatory frameworks.

## E. MACI Planner Algorithm

---
**Algorithm 1** $\mathcal{MP}$: Planner for Planning a Plan

---
**input** Objectives $\mathcal{O}$, explicit constraints $\mathcal{C_E}$, agent pool $\mathbf{A}$, people $\mathcal{P}$, metrics $\mathcal{M}$

**output** Optimized workflow $\mathbf{W}^* = (\mathcal{N}, \mathcal{E})$     **(Eq**. 1)

  // Phase 1: Network Construction

  1. Extract roles $\mathcal{N}$ from $\mathcal{O}$     **(Eq**. 2)

  2. Identify dependencies $\mathcal{E}$ from $\mathcal{C_E}$     **(Eq**. 3)

  // Phase 2: Agent Assignment

  3. Assign agents to nodes: $\forall n \in \mathcal{N}$, select $\alpha_n \in \mathbf{A_n}$ **(Eq**. 4)

  4. Assign agents to dependencies: $\forall e_{ij} \in \mathcal{E}$, select $\alpha_{ij} \in \mathbf{A_e}$     **(Eq**. 5)

  // Phase 3: Iterative Refinement

  **while** improvement in $V(\mathbf{W}, \mathcal{M})$ **do**

    **for all** $n \in \mathcal{N}$ **do**

      Update role-person mappings     $f_{\text{role}}(n, \mathcal{P})$

    **end for**

    **for all** $e \in \mathcal{E}$ **do**

      Verify dependencies via assigned edge agents

    **end for**

    **if** $V(\mathbf{W_{new}}, \mathcal{M}) > V(\mathbf{W_{current}}, \mathcal{M})$ **then**

      $\mathbf{W_{current}} \leftarrow \mathbf{W_{new}}$

    **end if**

  **end while**

  **return** $\mathbf{W}^* = \mathbf{W_{current}}$

---

## F. Traveling Salesman Problem Experiment

### F.1. General Problem Specification

The TSP requires finding the shortest possible route visiting N locations exactly once, returning to the start:

- Inputs: N locations, distance matrix D[N][N]
- Output: Optimal tour T minimizing total distance
- Constraints: Each location visited once, return to start

**Computational Complexity - Brute Force**   For N locations:

- Number of possible tours = (N-1)!/2
- Time complexity = $O(N!)$
- Space complexity = $O(N^2)$

**Solution Methods**

1. Exact Methods: Representative methods are *Branch and Bound* (Land & Doig, 1960), *Dynamic Programming* (Bellman, 1962), and *Integer Linear Programming* (Dantzig et al., 1954).

2. Heuristics: Methods include *Nearest Neighbor*, *Insertion Methods*, and *Christofides Algorithm* (3/2-approximation) (Christofides, 1976).

3. Meta-heuristics: This category includes *Genetic Algorithms* (Holland, 1992), *Simulated Annealing* (Kirkpatrick et al., 1983), and *Ant Colony Optimization* (Dorigo & Stützle, 2004).

### F.2. W*: MACI Generated Planner for TSP

**Node Components (N)**   For TSP with n locations:

$$N = \{n_{\text{route}}, n_{\text{dist}}, n_{\text{valid}}\}, \text{ where} \quad (8)$$

- $n_{\text{route}}$: Route generation role
- $n_{\text{dist}}$: Distance calculation role
- $n_{\text{valid}}$: Solution validation role

**Edge Dependencies (E)**

$$E = \{e_{\text{spatial}}, e_{\text{sequence}}, e_{\text{complete}}\} \text{ where} \quad (9)$$

- $e_{\text{spatial}}$: Distance constraints between locations
- $e_{\text{sequence}}$: Visit order constraints
- $e_{\text{complete}}$: Tour completion requirements

**Agent Assignments**

Node Agents ($A_n$):

- Route Generation Agent: Generates candidate tours
- Distance Calculator Agent: Computes tour lengths
- Solution Validator Agent: Verifies tour validity

Edge Agents ($A_e$):

- Spatial Constraint Agent: Monitors distance feasibility
- Sequence Monitor Agent: Ensures valid visit order
- Completion Checker Agent: Verifies tour completeness

**Algorithm Selection**  Based on the size of the problem, an algorithm is selected to balance performance trade-offs and mitigate the exponential computational cost of the brute-force method.

**Validation Function**

$$V(W, M) = \begin{cases} -\infty & \text{if constraints violated} \\ -\text{tour\_length} & \text{if tour valid} \end{cases}$$

(10)

**F.3. Experiments, From Small to Large N**

1. N=5: Establish ground truth via brute force
2. N=10,20,100: Test LLM performance degradation
3. Metrics:
   - Solution quality vs optimal
   - Computation attempts before giving up
   - Error recognition capability

F.3.1. SMALL CAMPUS TOUR (N=5)

Plan an optimal route for campus tour guide visiting 5 key locations:

- A: Admissions Office (start/end)
- B: Library
- C: Student Center
- D: Science Building
- E: Sports Complex

**Distance Matrix (minutes)**

$$D = \begin{bmatrix} 0 & 5 & 8 & 4 & 7 \\ 5 & 0 & 6 & 3 & 8 \\ 8 & 6 & 0 & 5 & 4 \\ 4 & 3 & 5 & 0 & 6 \\ 7 & 8 & 4 & 6 & 0 \end{bmatrix}$$

(11)

**Constraints**

- Tour starts/ends at Admissions (A)
- Each location visited exactly once
- Total possible routes: (5-1)!/2 = 12
- Optimal solution can be verified by hand

**3.1.1. $W^*$ Workflow Components**

**Node Components (N)**

- $n_{\text{route}}$: Generates permutations starting/ending at A.
- $n_{\text{dist}}$: Computes tour length using distance matrix $D$.
- $n_{\text{valid}}$: Checks tour validity (start/end at A, no repeats).

**Edge Dependencies (E)**

- $e_{\text{spatial}}$: Enforces distance constraints from $D$.
- $e_{\text{sequence}}$: Ensures visit order consistency.
- $e_{\text{complete}}$: Validates all 5 locations are visited.

**Agent Assignments**

- **Node Agents**:
  - Route Generation Agent (handles $n_{\text{route}}$)
  - Distance Calculator Agent (handles $n_{\text{dist}}$)
  - Solution Validator Agent (handles $n_{\text{valid}}$)
- **Edge Agents**:
  - Spatial Constraint Agent (enforces $e_{\text{spatial}}$)
  - Sequence Monitor Agent (enforces $e_{\text{sequence}}$)
  - Completion Checker Agent (enforces $e_{\text{complete}}$)

**Selected Algorithm**  Brute-force.

**Validation Function**

$$V(W, M) = \begin{cases} -\infty & \text{if constraints violated} \\ -\text{tour\_length} & \text{if tour valid} \end{cases}$$

**3.1.2. Solution Steps**

**Step 1: Problem Parsing**

- Input: 5 locations with distance matrix $D$.
- Initialize node/edge agents and constraints.

**Step 2: Route Generation ($n_{\text{route}}$ Agent)**

- Generate all valid permutations: $\frac{(5-1)!}{2} = 12$ routes.
- Example permutations:
  - $A \to D \to B \to C \to E \to A$
  - $A \to B \to D \to C \to E \to A$

**Step 3: Distance Calculation ($n_{\text{dist}}$ Agent)**

- Compute total time for each route using $D$.

**Step 4: Solution Validation ($n_{\text{valid}}$ Agent)**

- Check all routes for:
  - Start/end at A (e.g., invalid route: $A \to B \to C \to D \to E \to B$).
  - No duplicate visits.

**Step 5: Edge Agent Validation**

- Spatial Constraint Agent: Verify $D_{i,j}$ matches edge weights.
- Sequence Monitor Agent: Confirm no backtracking (e.g., $B \to D$ allowed; $D \to B$ invalid unless part of loop).

**Step 6: Apply Validation Function**

- Assign $V = -\infty$ to invalid routes.

- Assign $V = -$tour_length to valid routes.

- Identify minimal $V = -24$ (i.e., maximal tour length 24 mins).

### 3.1.3 Solution

Optimal tour time: $\boxed{24}$ minutes, achieved by three routes:

- $A \to D \to B \to C \to E \to A$
- $A \to B \to D \to C \to E \to A$
- $A \to E \to C \to B \to D \to A$

F.3.2. Large Campus Tour (N=10)

Plan an optimal route for a guided tour through 10 locations:

- **Locations**: A (Admissions), B (Library), C (Student Center), ..., J (Sports Complex)

- **Distance Matrix**: Asymmetric travel times (minutes)

$$
D = \begin{pmatrix}
0 & 12 & 8 & 15 & 9 & 14 & 7 & 11 & 10 & 6 \\
10 & 0 & 7 & 14 & 6 & 16 & 9 & 13 & 5 & 8 \\
9 & 5 & 0 & 11 & 8 & 12 & 10 & 7 & 15 & 4 \\
14 & 8 & 12 & 0 & 10 & 9 & 13 & 6 & 11 & 7 \\
7 & 13 & 6 & 9 & 0 & 8 & 5 & 12 & 14 & 10 \\
11 & 9 & 15 & 8 & 12 & 0 & 7 & 10 & 13 & 5 \\
5 & 7 & 10 & 6 & 11 & 9 & 0 & 8 & 12 & 15 \\
8 & 14 & 4 & 10 & 7 & 13 & 6 & 0 & 9 & 11 \\
12 & 6 & 9 & 7 & 15 & 10 & 8 & 5 & 0 & 14 \\
9 & 10 & 7 & 13 & 5 & 11 & 14 & 8 & 12 & 0
\end{pmatrix}
$$

**Algorithm Selection**    Based on the size of the problem, $\mathcal{MP}$ selected the *Ant Colony Optimization* (Dorigo & Stützle, 2004) algorithm to achieve at least a 4x speedup. For $N \geq 10$, an approximate method is recommended.

3.2.1. ACO Method

**Parameters**

- 100 ants, 50 iterations, and $\rho = 0.1$ evaporation
- $\alpha = 1$ (pheromone weight), and $\beta = 2$ (heuristic weight)

The termination criteria can be modified to stop the algorithm if no meaningful improvement is observed after $k$ consecutive iterations.

**Algorithm**

1: Initialize $\tau_{ij} \leftarrow 1.0$, $\eta_{ij} \leftarrow 1/D_{ij}$
2: **for** 50 iterations **do**
3:    **for all** 100 ants **do**
4:       Build tour using $P_{ij} = \frac{[\tau_{ij}]^1 [\eta_{ij}]^2}{\sum [\tau_{ik}]^1 [\eta_{ik}]^2}$
5:       Record tour length $L_k$
6:    **end for**
7:    Evaporate pheromones: $\tau_{ij} \leftarrow 0.9\tau_{ij}$
8:    Deposit pheromones: $\tau_{ij} \leftarrow \tau_{ij} + \sum \frac{10}{L_k}$
9:    Track best tour
10: **end for**

3.2.2. Performance Comparison

The goalpost is the optimal time of $\boxed{60}$ minutes. Table 8 compares six different configurations, and three of the six achieve the optimal answer. Although TSP is a relatively simple scheduling problem with just one actor and no parallel execution, the benefit of having $\mathcal{MP}$ to validate results is still helpful to Glaude and GPT4o.

When asked to solve the problem without $\mathcal{MP}$, Glaude and GPT4o initially chose brute force, then switched to an approximation method without thorough deliberation (or perhaps they did, but did not output their reasoning process). However, DeekSeek picked Held-Karp, a computationally expensive method, even more expensive than brute force, arguing that the absolute computation time for $N = 10$ is only 0.2 seconds. $\mathcal{MP}$ was more deliberate, opting for brute force when $N = 5$ and ACO for $N = 10$.

### F.4. TSP Experiment Conclusion

This simple task demonstrates that $\mathcal{MP}$ can be valuable for monitoring the execution process, validating the correctness of intermediate results, and suggesting more efficient algorithmic approaches.

# G. Experiment Details: Meta-Planning for the Thanksgiving Dinner Task

The problem statement remains consistent with Section 3, with $\mathbf{W}^*$ generated by $\mathcal{MP}$ to enhance constraints and dependencies. Planning performance is compared across four configurations: DeepSeek, GPT4o, DeepSeek + $\mathcal{MP}$, and GPT4o + $\mathcal{MP}$.

### G.1. Phase 1: Network Construction

G.1.1. Node (Role) Specifications

First, meta-planner $\mathcal{MP}$ extracts roles ($\mathcal{N}$) with their required qualifications:

- $n_{\text{cook}}$: capability to prepare dinner

- $n_{\text{driver1}}$: capability to drive, pick up from airport

- $n_{\text{driver2}}$: capability to drive, pick up grandma

- $n_{\text{supervisor}}$: capability to monitor oven

*Table 8.* Comparison of Planners and Their Performance Characteristics. $\mathcal{MP}$ provides validation to improve accuracy.

| Planner | Best Results | Algorithm | Iters. | Advantages | Limitations |
|---|---|---|---|---|---|
| Claude | $92 \to 66$ mins | Nearest Neighbor | 3 | Efficiently implements greedy heuristic approach | Makes data reading errors, compromising solution accuracy |
| GPT4o | $75 \to 68$ mins | Genetic | 3 | Identifies effective termination conditions | Unable to implement exact algorithms like Held-Karp |
| **DeepSeek** | 60 mins | Held-Karp | **1** | Implements optimal algorithm correctly | None observed for this problem size |
| $\mathcal{MP}$ + Claude | $66 \to 60$ mins | Ant Colony Optimization | 2 | Provides validation and suggests iteration increases for improvement | Requires external guidance for algorithm selection and parameter tuning |
| **MP + GPT4o** | 60 mins | Ant Colony Optimization | **1** | Achieves optimal solution with precise execution | Requires more computational resources with larger ant population and iteration count |
| **MP + DeepSeek** | 60 mins | Ant Colony Optimization | **1** | Combines efficient algorithm selection with optimal parameter tuning | None significant for given problem |

### G.1.2. EDGE (DEPENDENCY) SPECIFICATIONS

Next, $\mathcal{MP}$ identifies dependencies ($\mathcal{E}$) between roles:

$$\mathcal{E} = \{e_{\text{temporal}}, e_{\text{spatial}}, e_{\text{safety}}\} \quad (12)$$

The critical dependencies include:

- $e_{\text{temporal}}$: - Turkey (4 hours) must finish by 6:00 PM - Side dishes (2 hours) must finish by 6:00 PM - Airport pickups must align with landing times

- $e_{\text{spatial}}$: - Driver-passenger location matching - Travel time constraints between locations

- $e_{\text{safety}}$: - Continuous oven supervision requirement

### G.2. Phase 2: Agent Assignments

After constructing the network structure, $\mathcal{MP}$ selects and assigns agents to monitor both the roles and dependencies.

### G.2.1. NODE (ROLE) AGENT ASSIGNMENT

For each role, $\mathcal{MP}$ selects monitoring agents with the required capabilities:

$$f_{\text{role}} : \mathcal{N} \to \mathbf{A} \quad (13)$$

The role monitoring agents include:

- Cook Monitor: Tracks cooking timeline, coordinates meal components

- Driver Monitor: Validates driver availability

- Supervisor Monitor: Ensures oven supervision

- Resource Monitor: Manages vehicle assignments and actor schedules

### G.2.2. EDGE (DEPENDENCY) AGENT ASSIGNMENT

For the identified dependencies, $\mathcal{MP}$ assigns specialized monitoring agents:

$$f_{\text{edge}} : \mathcal{E} \to \mathbf{A} \quad (14)$$

Dependencies require these monitoring agents:

- Temporal Agent: Manages timing constraints (cooking durations, travel times, arrival schedules)

- Spatial Agent: Tracks location constraints (airport-home-grandma routes)

- Safety Agent: Ensures oven supervision constraint remains satisfied

The resulting agent assignments create a complete monitoring system where:

- Role agents track individual actor assignments and qualifications

- Edge agents monitor interactions and dependencies between roles

- All agents coordinate to maintain global constraint satisfaction

*Table 9.* Node (Role) Monitoring Agent Requirements

| Agent | Input Protocol | Output Protocol |
|---|---|---|
| Cook Monitor | Role: cook Qualifications: skills Time: prep and cook | Status: progress Alerts: timing issues! Updates: completed? |
| Driver Monitor | Role: driver Qs: license, rest Where: current GPS | Status: availability Alerts: fatigue warnings Updates: new GPS |
| Supervisor Monitor | Role: supervisor Location: house Duration: cover time | Status: covered? Alerts: coverage gaps! Updates: role transitions |

*Table 10.* Edge (Dependency) Monitoring Agent Requirements

| Agent | Input Protocol | Output Protocol |
|---|---|---|
| Temporal | Start times
Durations
Deadlines
Buffer requirements | Schedule conflicts
Timing violations
Schedule updates |
| Spatial | Locations
Routes
Travel times
Traffic conditions | Route violations
Location conflicts
Path updates |
| Safety | Critical constraints
Resource states
Coverage requirements | Safety violations
Resource conflicts
Mitigation plans |

### G.2.3. COMMON SENSE CONSTRAINT ANALYSIS (PERFORMED BY AN LLM)

A common sense agent identifies the following implicit constraints that can affect Thanksgiving dinner planning. This list is generated by Claude given the problem statement.

- *Physical Processing Times:*
  - Airport luggage claim: 30 minutes
  - Car rental procedures: 30 minutes
  - Holiday traffic variations
  - Winter weather considerations
- *Human Factors:*
  - Driver fatigue after long trips
  - Cooking preparation overhead
  - Multi-tasking limitations
  - Task switching delays
  - Required rest periods
- *Resource Dependencies:*
  - Vehicle passenger capacity
  - Oven temperature management
  - Kitchen workspace limits
  - Shared resource coordination
- *Social Considerations:*
  - Personal preferences for interactions
  - Family dynamics in assignments
  - Post-travel guest comfort
  - Host preparation requirements

### G.2.4. COMMON SENSE CONSTRAINT ANALYSIS AND VERIFICATION (HUMAN IN THE LOOP)

The common sense constraints identified above require different verification approaches:

**Agent-Required Information**  These constraints need specialized agents to verify and quantify:

- *Airport Operations*

- United Airlines' average luggage delivery time at BOS Terminal B
- Terminal B to rental car center: shuttle schedule, walking options
- Historical flight delay patterns for November at BOS
- *Weather and Traffic*
  - Boston weather forecast for the event date
  - Historical traffic patterns on Thanksgiving days
  - Impact on airport-city-suburb travel times
- *Task Dependencies*
  - Kitchen workflow analysis for parallel cooking tasks
  - Resource contention in meal preparation
  - Critical path identification in cooking timeline

**Human Verification**  Certain constraints require explicit human input to ensure that the planning process takes into account subtle interpersonal and individual considerations. These include:

- *Family Dynamics*
  - Preferred pickup arrangements for Grandma (e.g., Grandma loves to have a grandson surprise her).
  - Optimal relationship-based task pairings.
  - Social comfort factors in assignments (e.g., Sarah and Grandma do not work together in the kitchen).
- *Personal Capabilities*
  - Individual cooking experience levels.
  - Driver comfort with airport navigation.
  - Multi-tasking abilities of participants.

This separation ensures that agents focus on collecting quantifiable data while humans provide essential social and personal insights. $\mathcal{MP}$ can then integrate both types of information into the final workflow design.

### G.3. Agent Requirements and Assignments

The $\mathcal{MP}$ requires two categories of agents. $\mathcal{MP}$ specifies their requirements in the protocol buffer format in Table 9 for the nodes and Table 10 for the edges, respectively.

Each agent must implement these protocols to participate in the workflow. The meta-planner selects agents from the pool based on their ability to satisfy these interface requirements. During execution, agents communicate through these standardized protocols while maintaining their specialized monitoring functions.

### G.4. Monitoring Protocols and Dynamic Adjustments

The workflow monitoring operates through a hierarchical protocol system that enables both routine supervision and dynamic adjustments.

*Table 11.* Complete Workflow Specification: Nodes, Edges, and Agent Assignments

| Type | Component | Requirements | Agent Protocol | Dependencies |
|---|---|---|---|---|
| *Node Components (Roles)* | | | | |
| Node | Cook Role (Sarah) | - Turkey (4hr)
- Side dishes (2hr)
- Kitchen management
- Time management | Input: schedule, resources, recipes
Output: task progress, completion
Monitor: kitchen_state() → status
Validate: cooking_constraints() | Connected to:
- Supervisor
- Resource edges |
| Node | Driver1 (James/Michael) | - Valid license
- Airport navigation
- Car rental capable
- Rest state adequate | Input: flight times, routes
Output: location, ETA
Monitor: driver_state() → status
Validate: driver_constraints() | Connected to:
- Airport pickup
- Travel edges |
| Node | Driver2 (Flexible) | - Valid license
- Local navigation
- Availability window
- Rest state adequate | Input: pickup schedule, route
Output: location, ETA
Monitor: driver_state() → status
Validate: driver_constraints() | Connected to:
- Grandma pickup
- Travel edges |
| Node | Supervisor (Flexible) | - Home presence
- Oven monitoring
- Safety awareness
- Time commitment | Input: cooking schedule, rules
Output: supervision status
Monitor: safety_state() → status
Validate: safety_constraints() | Connected to:
- Cook role
- Safety edges |
| *Edge Components (Dependencies)* | | | | |
| Edge | Temporal | - Schedule tracking
- Buffer management
- Sequence logic
- Critical path | Input: timestamps, durations
Output: schedule conflicts
Monitor: schedule_state() → alerts
Optimize: timeline_adjust() | Connects:
- All roles
- All activities |
| Edge | Spatial | - Location tracking
- Route optimization
- Traffic updates
- Distance constraints | Input: locations, routes
Output: travel updates
Monitor: location_state() → alerts
Optimize: route_adjust() | Connects:
- Drivers
- Locations |
| Edge | Resource | - Vehicle allocation
- Kitchen resources
- People availability
- Capacity limits | Input: resource demands
Output: allocation status
Monitor: resource_state() → alerts
Optimize: resource_adjust() | Connects:
- All roles
- All resources |
| Edge | Safety | - Oven monitoring
- Driving safety
- Food safety
- Critical rules | Input: safety requirements
Output: violation alerts
Monitor: safety_state() → alerts
Enforce: safety_rules() | Connects:
- All roles
- Critical tasks |

**Basic Monitoring Protocol**   Each agent maintains a continuous monitoring cycle:

$$\text{monitor} : \text{State} \rightarrow \{\text{normal, warning, violation}\} \quad (15)$$

For example, the temporal agent tracks schedule adherence:

$$\Delta t = t_{\text{planned}} - t_{\text{actual}} \begin{cases} \text{normal} & \text{if } |\Delta t| < \text{buffer} \\ \text{warning} & \text{if buffer} \leq |\Delta t| < \tau \\ \text{violation} & \text{if } |\Delta t| \geq \text{ threshold } \tau \end{cases} \quad (16)$$

**Dynamic Adjustment Mechanism**   When deviations occur, the system initiates a three-phase response:

1. *Impact Assessment*:

$$\text{impact}(e) = \sum_{n \in \text{affected}(e)} \text{severity}(n) \times \text{urgency}(n) \quad (17)$$

2. *Solution Generation*:

$$S^* = \underset{s \in \text{Solutions}}{\arg \min} \{\text{cost}(s) | \text{feasible}(s)\} \quad (18)$$

3. *Coordination Protocol*:

$$\text{update} : (W_{\text{current}}, S^*) \rightarrow W_{\text{new}} \quad (19)$$

For instance, if James's flight is delayed:

- Spatial agent detects arrival time change
- Temporal agent calculates ripple effects
- Role agents evaluate reassignment options
- Safety agent verifies continued supervision coverage

The meta-planner $\mathcal{MP}$ coordinates these responses while maintaining global constraint satisfaction.

### G.5. Integrated Workflow Network

Table 11 presents the resulting workflow network $\mathbf{W}^*$, which includes all nodes and edges, and their assigned agents and protocols.

1. *Role Nodes:*
   - Cook1: Sarah (primary) or Grandma (if at home) with 4-hour turkey + 2-hour sides

- Driver1: James (after car rental) or Michael
- Driver2: Available person after initial pickups
- Supervisor: Must be present while turkey cooks

2. *Dependencies:*
   - Temporal: Verified airport processing + travel times
   - Spatial: Traveling routes with traffic consideration
   - Safety: Continuous oven supervision requirement

3. *Agent Monitoring:*
   - Temporal Agent: Schedules with verified buffer times
   - Spatial Agent: Real-time location and route mgmt.
   - Safety Agent: Role coverage for supervision

**G.6. Agent Interaction Specifications**

Please, see Table 12.

**G.7. New Problem Statement Revised with W*∗**

Given the $\mathbf{W}^*$ generated by MACI's meta-planner $\mathcal{MP}$, the Thanksgiving Dinner Planning problem statement stated at the beginning of this section is revised as follows:

*Initial Setup:*
- Mom (Sarah) is hosting Thanksgiving dinner at 6:00 PM in Boston. The following family members are traveling:
- Dad (James) flying from San Francisco, landing at 1:00 PM Eastern time.
- Sister (Emily) flying from Chicago, landing at 2:30 PM
- Brother (Michael) driving from New York, estimated arrival 3:00 PM at home
- Grandma is healthy and needs to be picked up from her home in suburban Boston

*Critical Dependencies:*
- James must rent a car after landing
- Emily must be picked up from airport, no other transportation options are allowed
- Turkey needs 4 hours to cook, someone must be in the house once turkey is in oven for safety
- Side dishes require 2 hours of preparation, which can overlap with turkey
- Travel time between home and Boston airport is one hour (one-way)
- Travel between Boston airport and grandma home is one hour (one-way)
- Travel between home and grandma home 30 minutes (one-way)

**\* New Dependencies:**
- The airport luggage pickup time after landing is 30 minutes.
- Renting a car takes 30 minutes.

- One person can simultaneously prepare turkey and side dishes.
- Grandma prefers Michael to pick her up, provided that it does not cause the dinner time delay.
- Grandma and Sarah prefer not to cook together in the kitchen.
- Traffic congestion is not factored into current planning.

*Planning Question Set:*
1. All tasks and dependencies must be strictly observed in the plan, or the plan fails.
2. Dinner time is strictly at 6:00 PM, all tasks must be completed by then (redundancy).
3. Account for the idle time of each person.
4. The schedule consists of three columns: time, task, and assigned person(s).

**G.8. Experiment #1: Sequential Planner**

Once after the original plan was revised by $\mathcal{MP}$ to include more specific details, clarify ambiguous explicit constraints, and define implicit constraints, the performance of the three LLMs used in the experiment improved significantly. When the augmented plan $\mathbf{W}^*$ was input into DeepSeek, GPT4o, and Claude, each model successfully generated a feasible plan within two to three iterations. (The case study in Section 3 shows that DeepSeek was confusing and GPT4o repeatedly committed constraint violations.)

G.8.1. RESULTS: DEEPSEEK WINS

Upon closer examination of the number of iterations required to produce a feasible plan, DeepSeek and Claude each required one revision (two iterations), while GPT4o required two revisions (three iterations). In terms of scheduling quality, measured by slack time, total driving distance, and load balance, DeepSeek (Table 13) outperformed both Claude (Table 15) and GPT4o (Table 14). DeepSeek optimized time and effort by scheduling James to wait at the airport for 30 minutes to pick up Emily. In contrast, Claude scheduled James to drive home and then return to the airport to pick up Emily, resulting in unnecessary travel. GPT4o assigned James to return home and scheduled Michael to first pick up Emily and then proceed to pick up Grandma, leading to a less balanced load. A better solution to reduce travel time would have been to schedule Michael to pick up Emily first and then drive with her to Grandma's home to pick up Grandma, allowing all three to return home together. This adjustment would save 30 minutes of driving time and improve Grandma's overall happiness to see both grandchildren.

*Table 12.* Agent Interaction Protocols and State Transitions

| Interaction Type | Protocol | State Transitions | Validation Rules |
|---|---|---|---|
| *Node-to-Node Interactions* | | | |
| Cook ↔ Supervisor | Protocol: cooking_handoff() 
 Message: (task, duration, reqs.) | States: prep → cooking → comp. 
 Trigger: task_state_change() | Validate: coverage() 
 Alert: coverage_gap() |
| Driver1 ↔ Driver2 | Protocol: pickup_handoff() 
 Message: (location, time, passenger) | States: available → enroute → comp. 
 Trigger: location_change() | Validate: timing_feasible() 
 Alert: schedule_conflict() |
| *Edge Agent Operations* | | | |
| Temporal Agent | Protocol: schedule_monitor() 
 Message: (event, time, dependencies) | States: scheduled → active → comp. 
 Trigger: time_milestone() | Validate: timing_feasible() 
 Alert: delay_impact() |
| Spatial Agent | Protocol: location_track() 
 Message: (actor, position, dest.) | States: idle → moving → arrived 
 Trigger: position_update() | Validate: route_feasible() 
 Alert: travel_delay() |

*Table 13.* DeepSeek's Plan, Two Iterations

| Time | Task | Assigned |
|---|---|---|
| 1:00 PM | James lands at Boston | James |
| 1:00–1:30 PM | James picks up luggage | James |
| 1:30–2:00 PM | James rents a car | James |
| 2:00 PM | Turkey in oven (4 hours; requires monitoring) | Sarah |
| 2:00–3:00 PM | James waits at airport (idle) | James |
| 2:30 PM | Emily lands at Boston | Emily |
| 2:30–3:00 PM | Emily waits for luggage | Emily |
| 3:00 PM | James picks up Emily | James |
| 3:00 PM | Michael arrives home | Michael |
| 3:00 PM | Michael departs to Grandma | Michael |
| 3:30 PM | Michael picks up Grandma | Michael |
| 3:30–4:00 PM | Michael drives back home with Grandma | Michael |
| 3:00–4:00 PM | James drives Emily home (airport to home: 1 hour) | James |
| 4:00 PM | James and Emily home | James |
| 4:00 PM | M. and Grandma home | Michael |
| 4:00–6:00 PM | Sarah prepares side dishes | Sarah |
| 6:00 PM | Thanksgiving dinner begins | All |

*Table 14.* GPT4o's Plan, Three Iterations

| Time | Task | Assigned |
|---|---|---|
| 1:00 PM | Land at BOS Airport | James |
| 1:00-1:30 PM | Luggage pickup | James |
| 1:30-2:00 PM | Rent car | James |
| 2:00 PM | Start turkey | Sarah |
| 2:00-3:00 PM | Drive home | James |
| 2:30 PM | Land at BOS Airport | Emily |
| 3:00 PM | Arrive home | Michael |
| 3:00-4:00 PM | Drive to airport, pick up Emily | Michael |
| 4:00-5:00 PM | Return home with Emily | Michael |
| 5:00-5:30 PM | Drive to Grandma's | Michael |
| 5:30-6:00 PM | Return with Grandma | Michael |
| 4:00-6:00 PM | Prepare side dishes | Sarah |
| 6:00 PM | Dinner served | All |

### G.8.2. OBSERVATIONS ON ERRORS

**Handling Long Dependencies** Complex scheduling problems reveal cascading errors when dependencies overlap. Key constraints, especially multi-factor dependencies, often get dropped during iterative problem-solving.
**Reason**: Cognitive load limits simultaneous constraint track-

*Table 15.* Claude's Plan, Two Iterations

| Time | Task | Assigned |
|---|---|---|
| 1:00 PM | Land at BOS Airport | James |
| 1:00-1:30 PM | Luggage pickup | James |
| 1:30-2:00 PM | Rent car | James |
| 2:00 PM | Start turkey | Sarah |
| 2:00-3:00 PM | Drive home | James |
| 2:30 PM | Land at BOS Airport | Emily |
| 3:00 PM | Arrive home | Michael |
| 3:00-4:00 PM | Drive to airport, pick up Emily | James |
| 4:00-5:00 PM | Return home with Emily | James |
| 4:30-5:00 PM | Drive to Grandma's | Michael |
| 5:00-5:30 PM | Return with Grandma | Michael |
| 4:00-6:00 PM | Prepare side dishes | Sarah |
| 6:00 PM | Dinner served | All |

ing, making exhaustive verification difficult in single passes.
**Solution Framework**:

- Isolate and enumerate atomic task dependencies.
- Verify global constraint satisfaction.
- Implement systematic conflict resolution.

**Stale Memory and Iterative Revisions** Iterative solutions can propagate errors due to partial constraint resets.
*Reason*: Over-reliance on previous solutions without full constraint re-evaluation leads to compounding errors.
**Relation to Gödel's Incompleteness**:

- Systems capable of arithmetic contain unprovable truths.
- Similarly, inherited errors hinder consistent solutions.
- Clean-state resets necessary for error prevention.

**Implementation Strategy** Reset to baseline state for each iteration, fully re-evaluating all constraints.
*Core Challenges*:

- Nested dependency management.
- Residual error prevention.
- Cross-iteration consistency.

### G.9. Experiment #2: Reactive Planner for Flight Delay

At 10:00 AM Eastern time, Sarah is notified that James's flight is delayed by three hours, with a new arrival time of

4:00 PM. Incorporating this unexpected delay, $\mathcal{MP}$ generates a reactive plan, $\mathbf{W^R}$.

**Early Information Agent Addition**   The meta-planner adds an early information agent to monitor upstream events:

$$f_{\text{early}} : \mathcal{E}_{\text{upstream}} \rightarrow \text{alerts} \qquad (20)$$

The agent's protocol is defined as:

*Table 16.* Early Information Agent Specification

| Component | Flight Monitor | Impact Analyzer |
|-----------|----------------|-----------------|
| **Input** | Flight status, departure logs, weather | Alert details, workflow dependencies |
| **Output** | Alert(event, severity, delay) | Replan(affected_nodes, time_window) |

This addition allows the workflow to initiate replanning at the earliest possible moment when upstream changes occur, significantly enhancing the system's proactive planning capability. Since none of the planned elements have been executed, this reactive planning effectively functions as proactive planning.

In this experiment, the problem statement remains unchanged apart from James's updated arrival time.

*Initial Setup (Updated at 10:00 AM):*

- Mom (Sarah) is hosting Thanksgiving dinner at 6:00 PM in Boston. The following family members are traveling:
- Dad (James) flying from San Francisco, landing at 4:00 PM Eastern time [UPDATED].
- Sister (Emily) flying from Chicago, landing at 2:30 PM
- Brother (Michael) driving from New York, estimated arrival 3:00 PM at home
- Grandma is healthy and needs to be picked up from her home in suburban Boston

*Critical Dependencies:*

- James must rent a car after landing
- Emily must be picked up from airport, no other transportation options are allowed
- Turkey needs 4 hours to cook, someone must be in the house once turkey is in oven for safety
- Side dishes require 2 hours of preparation, which can overlap with turkey
- Travel time between home and Boston airport is one hour (one-way)
- Travel between Boston airport and grandma home is one hour (one-way)
- Travel between home and grandma home 30 minutes (one-way)

* **New Dependencies:**

- The airport luggage pickup time after landing is 30 minutes.
- Renting a car takes 30 minutes.
- One person can simultaneously prepare turkey and side dishes.
- Grandma prefers Michael to pick her up, provided that it does not cause the dinner time delay.
- Grandma and Sarah prefer not to cook together in the kitchen.
- Traffic congestion is not factored into current planning.

*Planning Question Set:*

1. All tasks and dependencies must be strictly observed in the plan, or the plan fails.

2. Dinner time is strictly at 6:00 PM, all tasks must be completed by then (redundancy).

3. Account for the idle time of each person.

4. The schedule consists of three columns: time, task, and assigned person(s).

### G.9.1. RESULTS: DEEPSEEK WINS

None of the LLMs cannot react appropriately to this new event without clearing their context buffers. As explained in Appendix G.8.2, this limitation is evident. The key takeaway is that for future runtime frameworks, we must ensure infrastructure support for selectively invalidating stale constraints. If a workflow is already in execution, completed steps and assignments cannot be erased or altered. For example, in a stock-market investment plan, when pertinent news arrives, $\mathcal{MP}$ cannot revert completed nodes or resolved dependencies in $\mathbf{W^R}$. For now, we treat the reactive plan as a new plan, given that no steps have been realized in the real world by 10:00 AM.

Table 17 presents GPT4o's plan. There are three severe constraint violations. Unfortunately, when asked to identify violations, it answers none. Therefore, $\mathcal{MP}$ is stuck without a feasible plan.

Table 18 depicts Claude's plan. It violated a couple of constraints in the first two attempts, but these were minor. For instance, in the second trial, it planned for Michael's round trip to Grandma's home to take 30 minutes. However, the key is that Claude can recognize its own error and make corrections in the next iteration.

When asked to produce an alternate plan to reduce wait time and improve load balancing, as Michael can suffer from severe fatigue, an implicit constraint, Claude generates another feasible plan in Table 19. In this plan, James picks Emily instead of Michael. Emily has to wait for James' availability for two hours at the airport.

*Table 17.* GPT4o's Infeasible Plan. Fail to proceed.

| Time | Task | Assigned |
|---|---|---|
| 10 - 2:00 PM | Prep side dishes (2 hours, overlaps with turkey cooking later) | Sarah |
| **X**2:30 - 3:00 PM | Pick up Emily from the airport | Sarah |
| **X**3:00 - 4:00 PM | Prep turkey and place it in the oven (4-hour cooking time) | Sarah |
| 3:00 - 3:30 PM | Michael drives to pick up Grandma | Michael |
| 3:30 - 4:00 PM | Drive Grandma home | Michael |
| 4:00 - 4:30 PM | James lands and gets luggage | James |
| 4:30 - 5:00 PM | James rents a car | James |
| **X**5:00 - 5:30 PM | James drives home | James |
| 5:00 - 6:00 PM | Set the table and clean kitchen | Emily |
| 5:30 - 6:00 PM | Michael helps Grandma settle in and assists with final prep | Michael |

There are clearly other alternatives to improve the schedule and eliminate Emily's wait time, but none of the LLMs can figure that out. For example, using the time between 10:00 am and 1:00 pm, Sarah could have picked up Grandma and assigned her to cook, allowing Sarah to be available as a driver.

*Table 18.* Claude's Reactive Plan #1, Three Iterations

| Time | Task | Assigned |
|---|---|---|
| 13:00 | Start turkey in oven | Sarah |
| 14:00 | Start side dishes | Sarah |
| 14:30 | Land at airport | Emily |
| 15:00 | Arrive home from NY | Michael |
| 15:00 | Leave for airport | Michael |
| 16:00 | Land at airport | James |
| 16:00 | Pick up Emily | Michael |
| 17:00 | Arrive home with Emily | Michael |
| 17:00 | Leave for Grandma | Michael |
| 18:00 | Arrive home with Grandma | Michael |
| 18:00 | Arrive home | James |
| 18:00 | Dinner starts | All |

DeepSeek offers a clever alternative by routing Michael directly to the Boston airport without stopping at home first. This is a pleasant common-sense inference that the other two LLMs failed to include themselves. (This was supposed to be provided by $\mathcal{MP}$'s common-sense spatial reasoning, but it did not.)

However, Michael could drive to Grandma's home after picking up Emily. This schedule not only saves 30 minutes but also makes Grandma happy by allowing her to surprisingly see two grandchildren at the same time.

G.9.2. OBSERVATIONS ON ERRORS

The initial meta-planner failed to recognize a critical opportunity in early information detection. A flight delay from

*Table 19.* Claude's Reactive Plan #2. Michael can rest.

| Time | Task | Person |
|---|---|---|
| 13:00 | Start turkey | Sarah |
| 14:00 | Start side dishes | Sarah |
| 14:30 | Land at airport | Emily |
| 15:00 | Arrive from NY | Michael |
| 16:00 | Land at airport | James |
| 16:30 | Leave for Grandma | Michael |
| 16:30 | Get rental car | James |
| 17:00 | Pick up Emily | James |
| 17:00 | Pick up Grandma | Michael |
| 17:30 | Return home with Grandma | Michael |
| 18:00 | Arrive with Emily | James |
| 18:00 | Dinner starts | All |

*Table 20.* DeepSeek's Reactive Plan. Three Iterations. Routing Michael directly to BOS is smart.

| Time | Task | Assigned |
|---|---|---|
| 10:00 AM | Michael departs New York for Boston Airport (4-hour drive). | Michael |
| 2:00 PM | Start cooking turkey | Sarah |
| 2:30 PM | Emily lands at Boston | Emily |
| 3:00 PM | Emily gets her luggage | Emily |
| 3:00 PM | Michael arrives at Logan airport, picks up Emily. | Michael |
| 3:00–4:00 PM | Drive Emily home | Michael |
| 4:00 PM | Michael departs for Grandma | Michael |
| 4:00 PM | James lands at Boston Airport | James |
| 4:00–4:30 PM | James picks up luggage | James |
| 4:30–5:00 PM | James rents car (30 minutes). | James |
| 4:30 PM | Michael arrives at Grandma's | Michael |
| 5:00 PM | Michael & Grandma home. | Grandma |
| 5:00–6:00 PM | James drives home from BOS | James |
| 4:00–6:00 PM | Sarah prepares side dishes (overlaps with turkey). | Sarah |
| 6:00 PM | James arrives home. Dinner served. | All |

SFO to BOS becomes known at departure time (10:00 AM EST) rather than arrival time (1:00 PM EST). An early information agent could enable replanning three hours sooner by monitoring flight departures. To remedy this oversight, the meta-planner adds an early information agent specification, detailed in Table 16. DeepSeek was aware of this alert in a timely manner, but Claude was not.

**G.10. Conclusion**

Our concluding remark is that we may not be able to rely on LLMs alone to cover all constraints and react promptly to various alerts. This reinforces that the MACI architecture is on the right path to address all the aforementioned limitations of LLMs, some of which cannot be rectified.

