# OpenReview forum: "Position: Limitations of LLMs Can Be Overcome by Carefully Designed Multi-Agent Collaboration"
_ICML.cc/2025/Position_Paper_Track — Submitted to ICML 2025 Position Paper Track_

### Official Review · Reviewer_Rdci · 2025-03-10

**Significance:** 3
**Argument Clarity:** 3
**Rating:** 2
**Confidence:** 5

**Questions:**

-please address the concerns I raised in the weakness section of this review in above, i.e., comparative evaluation relative to existing multi-agent approaches, guarantee of correctness (e.g., feasibility of the the plan resulting from meta-planner that generated dependency graphs and constraint models), robustness to changing events, etc…

- can authors comments on instances that the framework will be limited to handle and will produce errors and failures? What mechanisms is in place to handle such cases?

-can authors also comment on the cost and benefit analysis of the proposed multi-agent approach planning vs a single agent planning framework together with some form of feasibility check. Can authors comments on large scale problem solving capabilities? What about optimality? For example, how the performance behave as the number of cities increases in the TSP problem. Do authors expect trade offs would be necessary.

Other questions:
-There is no info about which side of suburb grandma is living, so LLm cannot know if there is better route for Michael. Your criticism of suboptimality of the solution given by DeepSeek appears invalid.

-The assignment in equ 5 is not ensuring that the agent will satisfy the constraint. Just like the thanksgiving dinner example, these agents are all qualified to meet the constraint but they simply did not in the single LLm case. This will happen in the proposed system too. LLM can even ignore your instructions because of its overfitting bias.

-a diagram of the overall system would be very helpful. It is strange that the authors preferred to verbally describe everything without any illustration of how the components are interrelated.

**Discussion Potential:**

3

**Paper Summary:**

This paper is focused on shortcomings of Large Language Models (LLMs) when performing complex reasoning and planning tasks. the authors identify main limitations of existing works as: (1) LLMs are unable to verify their own reasoning reliably, (2) LLMs are not good at handling of long-term constraints due to their attention biases, and finally, (3) LLMs lack cannot properly use their common-sense knowledge in structured reasoning scenarios. To address these challenges, the authors propose the Multi-Agent Collaborative Intelligence (MACI) framework, enhancing reasoning/planning performance by systematically combining meta-planning with distributed validation among specialized agents.

MACI includes three main modules that collaborate with each other: (1) a Meta-Planner (MP), responsible for analyzing task requirements, identifying roles and constraints, and dynamically generating a dependency graph (or workflow template) tailored to the task.; (2) specialized reasoning agents whose roles are to handle domain-specific reasoning tasks; and (3) run-time monitor agent who handles realtime adjustments to the static plan in response to unexpected changes and  validation mechanisms ensuring adherence to constraints by cross-checking across multiple distributed agents. The paper considers couple of empirical evaluations, specifically, focusing on tasks such as the Traveling Salesperson Problem (TSP) and Thanksgiving dinner planning. The authors demonstrated that existing LLM architectures makes errors in satisfying certain constraints or provide suboptimal solutions when facing complex constraint management. The authors also show that the MACI framework improves the performance as such, the authors argue that multi-agent collaboration frameworks like MACI is critical for overcoming those shortcomings of existing single agent LLM-based reasoning systems.

**Position:**

Yes

**Position In Title:**

Yes

**Related Work:**

2

**Strengths And Weaknesses:**

Strength:
-The paper topic is very timely to the ICML community.

-The paper identifies three major limitations in current LLMs in planning and reasoning tasks: limitation in self-verification, limitation in handling of constraints due to attention biases, and limitation in use of common-sense reasoning in the tasks.

-MACI proposes a novel meta-planning strategy that manages/orchestrates the entire task among the components, separating high-level planning, constraint management, and detailed execution among various specialized reasoning agents. It has the advantage of being more robust and adaptive over single-LLM model approaches for planning.

-The paper performs empirical studies using realistic scenarios such as the Thanksgiving Dinner Planning task and Traveling Salesperson Problem (TSP). It illustrated existing single model LLM deficiencies. It also showed that the proposed MACI can improve upon those complex planning tasks.


Weakness:

-The paper may be more suitable as a regular contribution rather than a position paper, because multi-agent frameworks for improving LLM-based planning and reasoning have already been explored by multiple authors. Numerous approaches already exist that separate reasoning, validation, and execution across multiple agents or modules to enhance LLM effectiveness in planning/reasoning tasks. Thus, it remains unclear precisely what fundamental shift or conceptual departure this position paper introduces compared to existing multi-agent reasoning architectures. While the specific implementation details or individual agent roles might be novel, which makes it suitable for a regular paper, the work does not clearly establish as to how its proposed approach fundamentally differs from or substantially shifts from the view of current LLM based multi-agent methodologies.

-The evaluations in the paper only focused on empirical results. It is clear that the framework cannot offer theoretical guarantees regarding the correctness and feasibility  of MACI’s planning and constraint satisfaction. The proposed MACI framework still lacks guarantee of Correctness. A classical formal examination of MACI’s meta-planner can ensure its effectiveness and constraint satisfaction capabilities.

-it is unclear why this work should be viewed as a fundamental shift from existing works that also use both planner and verifier. The work needs more comparative analysis with related multi-agent approaches. The evaluations are only focused on single agent LLM models. But what about comparison with existing multi agent LLMs that are proposed for reasoning, or planning.

**Support:**

3

---

### Official Review · Reviewer_vq2Z · 2025-03-13

**Significance:** 2
**Argument Clarity:** 2
**Rating:** 1
**Confidence:** 3

**Questions:**

No

**Discussion Potential:**

2

**Paper Summary:**

This position paper argues that Large Language Models (LLMs) face inherent limitations in reasoning, self-validation, and constraint management, making them inadequate for complex planning tasks. The authors propose a Multi-Agent Collaborative Intelligence (MACI) framework as a solution, which consists of (1) a Meta-Planner (MP) that organizes tasks and constraints, (2) Specialized and Common Agents responsible for domain-specific and general-purpose reasoning, and (3) a Run-Time Monitor that dynamically adjusts plans in response to changing conditions. The paper presents experiments demonstrating MACI's superiority in structured planning scenarios, such as the Traveling Salesperson Problem (TSP) and a Thanksgiving dinner planning task, where single LLMs fail due to cognitive tunneling, local optimization bias, and lack of common sense reasoning. The authors claim that MACI represents a paradigm shift in AI planning, offering a scalable, adaptable, and verifiable alternative to monolithic LLMs.

**Position:**

No

**Position In Title:**

Yes

**Related Work:**

2

**Strengths And Weaknesses:**

**Strengths**

1. The paper provides a reasonable critique of LLMs’ deficiencies in self-validation, constraint handling, and logical consistency. The discussion on Gödel’s incompleteness theorem as a theoretical limitation is thought-provoking, even if somewhat overstated.

2. The MACI framework is well-articulated, with clear components designed to address different aspects of planning and reasoning. The distinction between meta-planning, task-specific agents, and runtime monitoring adds clarity to the framework.

3. The Thanksgiving dinner scheduling experiment effectively highlights common LLM mistakes in sequential and reactive planning, reinforcing the paper’s argument that unstructured LLM outputs are unreliable for complex workflows.

4. The discussion on how MACI differs from existing multi-agent frameworks (e.g., AutoGen, CrewAI, CAMEL) is useful, highlighting the need for real-time adaptability and improved global coordination.

**Weaknesses**

1. The idea of decomposing tasks into specialized agents with validation mechanisms is not new. Similar concepts exist in automated planning (e.g., hierarchical task networks), multi-agent reinforcement learning, and blackboard architectures. The authors fail to clarify what fundamentally sets MACI apart from prior systems.

2. The paper repeatedly emphasizes that LLMs cannot self-validate due to Gödelian constraints, but this is a philosophically weak analogy. Self-verification can be improved via external calibration, Bayesian reasoning, or structured output validation—solutions not discussed.

3. The evaluation consists of toy problems (TSP, Thanksgiving dinner planning), which do not convincingly prove MACI’s real-world impact. The paper lacks large-scale experiments on industrial-scale planning or multi-step reasoning benchmarks.

4. While MACI is presented as a theoretically superior framework, the authors do not discuss practical challenges such as computational cost, communication overhead, or integration feasibility with existing LLM architectures.

5. The paper weakly addresses counterarguments. It dismisses single LLM improvements and multi-agent systems without quantitative comparisons or alternative baselines beyond basic scheduling tasks.

6. The excessive reliance on Gödel’s incompleteness theorem and formal systems makes the argument feel unnecessarily abstract. The practical limitations of LLMs (e.g., hallucination, limited memory, and data bias) are well-documented, but the authors over-intellectualize the issue without offering better empirical grounding.

**Support:**

2

---

### Official Review · Reviewer_Kjny · 2025-03-18

**Significance:** 3
**Argument Clarity:** 2
**Rating:** 2
**Confidence:** 4

**Questions:**

Could the authors elaborate on how MACI quantitatively and qualitatively compares with existing multi-agent frameworks (e.g., Multi-LLM Debate, CAMEL) in terms of performance and adaptability in dynamic planning scenarios?

**Discussion Potential:**

3

**Paper Summary:**

This paper introduces a framework called Multi-Agent Collaborative Intelligence (MACI) that aims to overcome three fundamental limitations observed in large language models (LLMs) when handling complex planning tasks: reliance on pattern matching over deliberate reasoning, lack of self-validation (akin to issues highlighted by Gödel’s incompleteness), and inconsistent constraint management. The authors propose a structured multi-agent architecture—comprising a meta-planner, domain-specific and general-purpose agents, and a run-time monitor—that decomposes planning problems into specialized sub-tasks and coordinates their interaction. The paper validates the framework through two experimental case studies, namely Thanksgiving dinner planning and the Traveling Salesman Problem (TSP), demonstrating improvements in planning consistency and adaptability.

**Position:**

No

**Position In Title:**

Yes

**Related Work:**

3

**Strengths And Weaknesses:**

## Strengths
- **Detailed Architectural Design**: The authors offer a comprehensive description of the MACI framework, including its meta-planner, specialized and common agents, and run-time monitoring mechanisms. This detailed exposition facilitates a clear understanding of the proposed system.
- **Empirical Validation**: The experimental validations using both a controlled planning scenario (Thanksgiving dinner planning) and a well-known combinatorial optimization problem (TSP) reinforces the practical relevance of the framework, highlighting its ability to manage dynamic constraints and improve planning consistency.

## Weaknesses
- **Scalability and Complexity Concerns**: Although the architecture is well-detailed, the system’s complexity raises questions regarding its scalability in real-world, high-dimensional planning scenarios.
- **Lack of A Broad Evaluations for Supporting the Position**: Despite of the effectiveness shown on TSP and multi-layered dinner planning problem, I think that to convincingly support the meta-level position the authors proposed, the current experimental results are still insufficient. How about the system’s performance under noisy or incomplete information.
- **Mismatch with Position Paper Requirements**: The biggest issue in my opinion is that this work seems not aligned with the aim or the requirements of Position Paper Track. This paper presents a complete method along with experimental validation. This approach aligns more closely with the main track’s focus on reporting original research and novel results, rather than advocating a broad perspective or stimulating discussion on high-level conceptual issues. A position paper should ideally concentrate on arguing a viewpoint or offering a *meta-level* discussion without necessarily including full experimental validations.

**Support:**

2

---

### Decision · Program_Chairs · 2025-04-27

**Decision:**

Reject

**Comment:**

The reviewers had concerns about both the relevance of this paper to position paper track, and also the specificity of the position

The authors didn't engage in a rebuttal.